# Reversed Mirror Therapy (REMIT) after Stroke—A Proof-of-Concept Study

**DOI:** 10.3390/brainsci13060847

**Published:** 2023-05-24

**Authors:** Luigi Tesio, Antonio Caronni, Cristina Russo, Giorgio Felisari, Elisabetta Banco, Anna Simone, Stefano Scarano, Nadia Bolognini

**Affiliations:** 1Department of Neurorehabilitation Sciences, Istituto Auxologico Italiano, IRCCS, Ospedale San Luca, 20149 Milano, Italy; 2Department of Biomedical Sciences for Health, University of Milan, 20133 Milano, Italy; 3Department of Psychology, University of Milano-Bicocca, 20126 Milano, Italy; 4Neuropsychological Laboratory, Istituto Auxologico Italiano, IRCCS, 20122 Milano, Italy

**Keywords:** stroke, upper limb mobility, neurological rehabilitation, mirror movement therapies, mirror neurons, body image, crossmodal illusions

## Abstract

In mirror training (MIT), stroke patients strive to move their hands while looking at the reflected image of the unaffected one. The recruitment of the mirror neurons and visual-proprioceptive conflict are expected to facilitate the paretic voluntary movement. Here, a reversed MIT (REMIT) is presented, which requires moving hands while looking at the reflected image of the paretic one, giving the illusion of being unable to move the unimpaired hand. This study compares MIT and REMIT on post-stroke upper-limb recovery to gain clues on the mechanism of action of mirror therapies. Eight chronic stroke patients underwent two weeks of MIT and REMIT (five sessions each) in a crossover design. Upper-limb Fugl-Meyer, Box and Block and handgrip strength tests were administered at baseline and treatments end. The strength of the mirror illusion was evaluated after each session. MIT induced a larger illusory effect. The Fugl-Meyer score improved to the same extent after both treatments. No changes occurred in the Box and Block and the handgrip tests. REMIT and MIT were equally effective on upper-limb dexterity, challenging the exclusive role of mirror neurons. Contrasting learned nonuse through an intersensory conflict might provide the rationale for both forms of mirror-based rehabilitation after stroke.

## 1. Introduction

Post-stroke upper limb recovery in adults remains unsatisfactory [1,2]. In recent decades, the idea emerged that paresis could reflect a maladaptive acquired behaviour, worsening the brain injury’s direct effects. In particular, the paresis would partly reflect a form of “learned nonuse”, which, in principle, might be the target of behavioural treatments [3].

The term learned nonuse (or, more appropriately, “acquired nonuse”, given that no veritable learning seems involved [4]) indicates the reduced spontaneous use of the affected upper limb to its fullest potential, commonly associated with the overreliance on the unaffected one [5]. Stroke patients are affected by learned nonuse when they avoid using their impaired extremities, even though the motor circuitries directed to the diseased limbs are (at least partly) spared. To cite a common impression in the stroke recovery clinic: “he can, but does he?” [6].

It is believed that learned nonuse stems from the impairment of the sensorimotor cortices of the lesioned hemisphere [7]. It is well known that the excitability of the spared corticospinal fibres of the paretic limb muscles is depressed after damage to the primary motor cortex [8]. Limb immobility also depresses corticospinal excitability [9]. Therefore, nonuse can further inhibit corticospinal tract activation in stroke. Moreover, compensatory hyper-reliance on the good limb also drives bi-hemispheric maladaptive changes in the motor network. For example, skill training with the unaffected upper limb may result in aberrant synaptogenesis, true maladaptive plasticity, on the perilesional motor cortex, hindering the paretic limb recovery [10]. In short, an unbalanced interhemispheric inhibition makes the deficit worse. Inhibiting the healthy limb (or even the healthy hemisphere in stroke, e.g., through non-invasive brain stimulation [11]) can be a rational therapeutic approach [7].

The learned nonuse behaviour is a general adaptive phenomenon. For instance, it underlies amblyopia in strabismus [12] and limping after unilateral lower limb impairments (also in orthopaedic conditions) [13,14]. For the unilaterally affected subject, relying on the sound body side may allow a faster (often sub-optimal) recovery, much before the uncertain recovery of the affected side occurs [12].

Two therapeutic approaches can be proposed to counteract learned nonuse. A classic one is based on intensive motor training of the paretic limb while preventing compensation by the unaffected limb (e.g., the constraint-induced movement therapy—CIMT) [15]. The other approach is based on various forms of action observation, such as mirror training (MIT). Regarding this latter, there is a growing body of evidence, including systematic reviews, pointing out that MIT could effectively improve upper limb dexterity after a stroke [16,17,18]. The neural correlate of learning through observation has been postulated to be the recruitment of the mirror neuron system, i.e., cortical motor neurons activated by action observation [19]. After a stroke, these neurons would facilitate the voluntary recruitment of the spared yet inhibited corticospinal route from the lesioned hemisphere.

In MIT, a particular action observation form is applied. A mirror is placed between the two upper limbs in the mid-sagittal plane. In the classic paradigm, the patient is asked to move both upper limbs while looking at the reflected image of their unaffected limb. Thanks to the mirror neuron system, observation of a moving limb would foster re-learning of the observed movement.

The MIT adds to action observation an inter-sensory conflict between visual and proprioceptive inputs (vision signals normal motion of the affected upper limb; proprioception signals poor or no movement). The solution to the conflict would be moving the paretic limb. Suppose the intersensory conflict drives functional recovery or, at least, an increased motor output. In that case, a “reversed” MIT (REMIT, in which patients look at the reflected image of their impaired limb) might also be worth testing, although “action observation” may not help. As a mechanism of action, the “reversed” conflict between normal proprioceptive feedback and abnormal visual feedback can only be claimed.

The present work compares REMIT to the conventional MIT on upper limb hemiparesis in chronic post-stroke patients, shedding new light on the role of intersensory conflicts in mirror-based treatments.

## 2. Materials and Methods

This is a proof-of-concept, randomised, single-blind, double-crossover study. Participants provided their written informed consent to participate in this study. The study protocol followed the 2013 Declaration of Helsinki and was approved by the IRCCS Istituto Auxologico Italiano Ethics Committee (approval code: 2011_02_03_17).

Patient enrolment lasted two and a half years and was run at the Department of Neurorehabilitation Sciences–Ospedale San Luca of the Istituto Auxologico Italiano. Participants, all recruited by one of the authors (GF), were naïve to mirror therapies.

### 2.1. Participants

Patients who received rehabilitative treatments in our Department in the years before this study onset were contacted and offered to participate. Participants were consecutively enrolled.

Stroke outpatients with upper-limb motor deficits were recruited according to the following inclusion criteria: (1) age between 35 and 80 years; (2) right-handedness Edinburgh inventory >11/20 [20]; (3) first-ever ischemic or haemorrhagic cerebrovascular accident; (4) frontal or frontoparietal, cortico-subcortical unilateral lesion assessed by CT or MRI; (5) chronic phase of illness (≥6 months); (6) unmodified pharmacological therapy in the previous two months and at least four months from the latest botulin toxin treatment; (7) Brunnström stage = 3/4 [21]; (8) affected upper limb function ranging from 20 to 40% of the score of the unaffected upper limb function in the following subtests: Fugl-Meyer assessment, upper limb [22] (ul-FM); Box and Block [23]; hand grip strength [24] (see below for details on the behavioural tests).

Exclusion criteria were: (1) denial of informed consent; (2) history of subarachnoid haemorrhage or coma or neurologic diseases other than post-stroke hemiplegia; (3) unilateral spatial neglect, aphasia, or limb apraxia, assessed with a neuropsychological test battery; (4) campimetry deficit preventing full vision in the mirror or visual acuity < 8/10 (glasses allowed); (5) psychiatric disorders; (6) participation within the previous four months in a rehabilitation program with non-invasive brain stimulation or constraint-induced movement therapy; (7) contraindications to transcranial magnetic stimulation (TMS) [25].

### 2.2. Experimental Procedures

MIT and REMIT were administered to all patients in 5 daily one-on-one sessions (Monday to Friday) led by a trained physiotherapist. The therapist administering the mirror training was not engaged in patient assessment or data analysis. After a 2-day pause, the treatments were alternated (double crossover).

During the study period, participants only received mirror therapies. In other words, mirror therapies were not associated with other forms of therapeutic exercise.

Coin tossing was used to randomly select the treatment (MIT or REMIT) for the first enrolled patient. Then, the other treatments were assigned in an alternate sequence to the following patients to keep the first of the two treatments balanced between groups.

The MIT treatment replicated the original setting [26]. Patients sat at a table with their arms on the desk at the two sides of a mirror aligned with their body midline. The paretic arm was behind the non-reflective surface.

In this study, participants practised 19 movements according to the Fugl-Meyer assessment principle (see below):-Exercise 1: shoulder flexion and extension;-Exercise 2: shoulder abduction and adduction;-Exercise 3: elbow flexion and extension;-Exercise 4: forearm pronation and supination;-Exercise 5: forearm displacement on the table;-Exercise 6: shoulder internal and external rotation;-Exercise 7: hand from table to ear;-Exercise 8: forearm pronation and supination on the table;-Exercise 9: wrist abduction and adduction;-Exercise 10: wrist flexion and extension;-Exercise 11: thumb abduction and adduction;-Exercise 12: wrist flexion and extension with the prone hand;-Exercise 13: fingers flexion and extension;-Exercise 14: three knocks on the table;-Exercise 15: precision grip;-Exercise 16: finger purse supinated;-Exercise 17: finger purse on the side;-Exercise 18: 2nd finger extension and flexion;-Exercise 19: single fingers extension and flexion.

Videos of the 19 exercises are provided as Appendix A. The participants were asked to perform motions while moving the paretic hand at their best and watching the reflected (unaffected) hand in the mirror. For REMIT, patients were asked to keep the unaffected upper limb behind the mirror (Appendix A).

Exercises were taught by the therapist using verbal instructions, performing and showing the proper movements and assisting the participant in the first exercise repetitions. No additional material (e.g., illustrative sheets) was used.

As customary with therapeutic exercise, physical and cognitive fatigue was considered during mirror therapies administering. Fatigue was not of concern during treatments, even if this side effect was not recorded in this study with dedicated measures.

### 2.3. Clinical Assessment

The following tests were administered before the baseline (T0):NIH stroke scale [27], a 15 items scale, quantifies the neurologic severity of the syndrome caused by the stroke. The total score may range from 0 to 42 (the higher, the worse).Bamford classification [28]; based on the neurological symptoms, stroke is classified as anterior circulation stroke (total or partial), lacunar or posterior circulation stroke.Brunnström staging [21], a single-item scale to measure the severity of the hemiparesis (1 = flaccidity; 2–5 = weak/synergic movements; 6 = regular movements).Measures of corticospinal functioning obtained with TMS.

The corticospinal tract excitability was assessed bilaterally with TMS (Magstim^®^ 200², Magstim Company Ltd., West Wales, UK). Motor-evoked potentials (MEPs) were elicited in the resting left and right first dorsal interosseous (FDI) muscles by stimulating the contralateral motor cortices with a 70 mm figure-of-eight coil. The two hands were tested in a pseudorandom order (see Methods above). The coil was held tangential to the skull and, as customary for the stimulation of the hand area of the primary motor cortex, with its handle rotated 45° outward. MEPs were recorded with silver/silver chloride surface electrodes placed in a tendon–belly arrangement. The EMG signals were bandpass-filtered (50–1000 Hz), digitised (sampling rate 2 kHz) and stored on a computer for offline analysis (Synergy NCS EMG EO IOM System; Viasys Healthcare, Old Working, Surrey, UK).

On stimulation of both hemispheres, the resting motor threshold (RMT) was determined, i.e., the lowest TMS intensity necessary to evoke 3 out of 5 MEPs with peak-to-peak amplitude of at least 50 μV. In addition, the excitability of the primary motor cortex, defined as the mean amplitude of three MEPs evoked by TMS at 1.1 RMT, was measured [29].

The cortical representation of both FDI muscles was determined through TMS brain mapping [30]. Scalp positions to be stimulated were spaced 1.5 cm apart and arranged in two 10 × 10 grids referred to as the head vertex, one for each hemisphere. The position of the grid nodes was digitised with an optoelectronic neuronavigation system (SofTaxic Optic, EMS, Bologna, Italy), which then guided the TMS coil positioning on the grid’s stimulation sites. The TMS intensity for the mapping procedure was set at 1.1 of the RMT. In pseudorandom order, three TMS stimuli (pulse interval: 4–6 s) were delivered at each scalp stimulation site. If no MEP > 50 μV could be evoked from a stimulation point, that point was considered unexcitable. The map of excitable locations gave the FDI cortical representation area.

### 2.4. MIT and REMIT Effects: Behavioural Outcomes

Patients were assessed in three sessions (baseline—T0, at the end of the first week of treatment—T1 and at the end of the second week of therapy—T2) with the following instruments.

Fugl-Meyer assessment of the upper limb (ul-FM) [31] evaluates the capacity to complete isolated movements of the shoulder, elbow, wrist and fingers and multi-joint movements, different from the pathological synergisms often preserved after stroke [22,32,33]. The original scale includes 33 items scored 0-2 or 0-1-2, the higher the better the performance. For the current analysis, ul-FM scores were turned into interval measures running a Rasch analysis [34,35,36] with items’ “difficulty” calibrations from a previous study [37]. Several research groups reported the calibration of the ul-FM items with the Rasch analysis. The calibration provided in [37] is used here since, to our knowledge, this is the only study assessing the stability of items’ calibration over time. Ideally, the hierarchy of difficulty of the items should stay the same between time points [38]. Of note, as suggested by the literature [37], items 1, 2 and 18 (assessing the excitability of upper limb tendon reflexes) were not considered in the interval measure calculation since they reflect a construct different from the voluntary movement. Rasch analysis adopts logit units, unfamiliar to most health care professionals. For this reason, logit measures were converted here into a 0–100 scale (the higher, the better the condition), with 0 and 100 being assigned the lowest and the highest logit measure achieved in the calibration sample, respectively [37].Hand grip strength (HGS) [39] was measured by a dynamometer (Jamar Hydraulic dynamometer, Lafayette instruments, Lafayette, IN, USA). Patients are instructed to grasp the dynamometer handle as hard as possible and hold it for four seconds. The average peak force (kg) across three measurements is taken as the measure of grip strength.In the Box and Block test (BB) [23], 100 cubic wooden blocks (side 2.54 cm) are placed on one side of a box, split by a partition wall 15.2 cm high. Then, using the upper limb of the side where blocks are stored, the participants grasp and release to the opposite side of the partition, one block at a time, as many blocks as possible within one minute. The number of blocks displaced is counted.ABILHAND [40] is a cumulative questionnaire that measures the patient’s perceived difficulty performing 23 manual activities (e.g., washing hands or hammering a nail). In the original questionnaire, patients should only score the activities they performed during the latest three months. However, because of weekly assessments, participants were asked to fill out the questionnaire referring to the previous week for the current study. Thanks to published calibrations of item difficulty levels [41], the analysis could transform the total expected scores into linear “logit” measures of subjects’ “ability”. In this study, ABILHAND logit measures were transformed into more familiar 0–100 measures (higher score assigned to a better condition, the same procedure adopted here for the ul-FM measures) [42].

At the end of each daily session, participants also filled out the Mirror Illusion Questionnaire (MIQ, adapted from [43]) to measure the strength of the illusory perception brought about by MIT and REMIT. The MIQ comprised the following items: (1) “It felt like I was looking directly at my hand rather than at a mirror image”; (2) “It felt like both my hands were moving simultaneously”; (3) “It felt like the movements in the mirror were the same I was performing”; (4) “It felt like I was moving with less difficulty”; (5) “It felt like the hand I was looking at was my left/right hand”. Participants rated each item using a Likert scale of 1 to 7 (1 = strongly disagree to 7 = strongly agree), so the higher the total score, the stronger the illusory effect.

### 2.5. Statistics

Due to the restricted sample size, the median and the first to third quartiles range (Q1–Q3) were adopted as central tendency and dispersion indexes. Median and 95% confidence intervals (95% CI) were calculated with a bootstrapping procedure (10^4^ replications, random sampling with replacement).

According to the restricted randomisation procedure described above, patients were switched to the second after the first treatment (i.e., MIT or REMIT). Regression with “Group” (MIT-first or REMIT-first), “Session” (T0, T1 and T2) and their interaction as predictors were adopted. However, nonparametric statistics were preferred for hypothesis testing because of the small sample size. Thus, the Aligned Rank Transform (ART) ANOVA [44] was applied.

ART-ANOVA allows the nonparametric analyses of factorial datasets with repeated measures and interactions, such as those arising from crossover studies. Compared to parametric ANOVA, ART-ANOVA does not rely on the assumption of normally distributed and homogeneous residuals. Furthermore, compared to ANOVA calculated on ranks, which produces inaccurate results for interaction effects, ART-ANOVA estimates both main and interaction effects with appropriate power and type I error probability.

ART-ANOVA is a two steps procedure. First, responses are aligned for each predictor and interaction, and their mid-ranks are calculated (properly, the ART procedure). Second, the conventional ANOVA on the aligned ranks is conducted. It is worth stressing that ART-ANOVA is used in the circumstances similar to parametric ANOVA and interpreted as in ANOVA. Here, ART-ANOVA was run with “Group” as a between-subjects factor and “Session” as a within-subjects factor. Four distinct models were tested with the ul-FM, BB, HGS and ABILHAND as the predicted variables along with the three sessions. For the MIQ data (predicted variable), which were collected daily, ART-ANOVA was run with the following factors: “Week” (the first or second week of treatment) and “Treatment” (MIT vs. REMIT). In addition, the Week × Treatment interaction was also tested.

ART contrasts, i.e., contrast on responses aligned and mid-ranked on the contrasts’ factors, were subjected to post hoc testing [45].

A principal component analysis (PCA) was run to evaluate the association between the change in the upper limb motion measures and the neurophysiological measures recorded at baseline. This solution was preferred to a traditional correlation matrix to keep the number of significance tests low. Only principal components (PCs) with eigenvalue > 1 and the variables strongly correlated with these PCs (i.e., the variables with a correlation coefficient outside the −0.6 to 0.6 range) were retained. The significance of the correlation between the measured variables and the PCs was eventually calculated [46].

Type I error probability was set at 0.05, and the Holm correction for multiplicity [47] was applied to the post hoc tests.

R version 3.6.2 was used for statistics and graphics. Winsteps version 4.4.8 was used for turning the ABILHAND and ul-FM scores into the interval measures with item anchors provided by [41] and [48], respectively.

MRIcro software (v. 1.0) [49] was used to manually draw the cerebrovascular lesion on a standard MRI template (1 mm slice distance; voxels of 1 mm^3^).

## 3. Results

Table 1 summarises the demographic and clinical characteristics of the eight patients enrolled in this study.

Figure 1 shows, for each stroke participant, the extent of the cerebrovascular lesion(median lesion volume: 77 mm^3^, Q1–Q3 = 10.6–155.1 mm^3^). Regions of Interest (ROIs) defined the location and the size of the lesion for each patient as assessed with a standard MRI or CT performed for diagnosis purposes (see Section 2.1). ROIs were mapped by means of a template technique by manually drawing the lesion on the standard template from the Montreal Neurological Institute, on each 2D slice of a 3D volume [49]. Five patients showed left-sided brain lesions, and three showed a lesion to the right hemisphere. The primary motor cortex or the underlying white matter was lesioned in all patients.

Only one patient (P4) suffered a haemorrhagic stroke, while the remaining seven had an ischaemic one. For this patient, the stroke occurred 30 months before this study enrolment. Consistent with this delay between the vascular accident and the recruitment, no more bleeding was present at the MRI scan. In addition, the site and size of patient’s lesion was similar to that of the ischemic patients P2 and P7, involving the basal ganglia and the internal capsule.

In the four patients with MEPs in the paretic hand (Table 2), the cortical representation of the two FDI muscles was not symmetric. However, these results were inconsistent, since the FDI representation was reduced in the lesioned hemisphere in two patients and more minor in the intact one in the other two. Stroke participants suffered from severe to moderate upper limb impairment (ul-FM measure at T0 ranged from 19.16 to 59.16 out of a 0–100 potential range, see Table 1).

Cortical excitability data at T0 are summarised in Table 2. No MEPs in the paretic FDI could be found in four out of eight patients (P3, P4, P5, P6), even at the maximum stimulator output. By contrast, all patients had MEPs in the FDI of the unaffected hand. Furthermore, in the four patients showing MEPs in the impaired hand, the RMT was invariably higher when stimulating the affected, compared to the unaffected hemisphere. Moreover, MEPs evoked from FDI at a stimulation intensity of 1.1 RMT were smaller on the paretic than on the unaffected side.

### 3.1. Treatment Effects: Behavioural Outcomes

Changes in behavioural tests are represented in Figure 2, and their statistical analysis is in Table 3. In both groups, improvements emerged in ul-FM (Figure 2A) but not in BB, HGS and ABILHAND tests.

Regarding the ul-FM, the ART-ANOVA showed a main effect of Session (F_2,12_ = 6.23, *p* = 0.01), while the Group main effect (F_1,6_ = 0.03, *p* = 0.86) and the Session × Group interaction (F_2,12_ = 0.54, *p* = 0.60) were not significant. Post hoc testing showed that the ul-FM score was significantly larger (*p* = 0.03) at T1 (median: 47.90; Q1–Q3: 35.34–60.11) and T2 (48.23; 33.81–58.94), compared to T0 (44.30; 33.47–55.25). No difference was found between the T1 and T2 assessments.

The main effects and interaction of the models with BB, HGS and ABILHAND as predictors did not attain the significance level (Table 3).

Concerning the MIQ score (Figure 3), patients reported higher illusory effects after MIT, compared to REMIT (Treatment: F_1,70_ = 10.05, *p* < 0.01; Week: F_1,70_ = 1.56, *p* = 0.22; Week × Treatment: F_1,6_ = 0.01, *p* = 0.92).

### 3.2. Association between Motor Cortex Excitability and Post-Treatment Improvements

The PCA of the six TMS measures and the measured change at the ul-FM (delta T1-T0 and delta T2-T0) returned two principal components accounting for 75.6% of the total variance (PC1 and PC2, respectively). For PC1, a negative correlation was found as follows:(a)With the first and second ul-FM differences between time points (delta) (delta T1-T0: r = −0.87, *p*-value = 0.01; delta T2-T0: −0.94, *p*-value < 0.01);(b)Between the RMT of the lesioned hemisphere and PC1 (r = −0.63), although not significant (*p*-value = 0.10).

A positive correlation was found with the area of the FDI map of the sound hemisphere (r = 0.79, *p* = 0.04).

For PC2, a negative correlation was found for the area of the paretic FDI, and the MEP amplitude elicited at 1.1 times the RMT in the paretic FDI (−0.96 and −0.72, respectively). A positive correlation was found for the RMT of the paretic FDI (r = 0.67). However, the only significant correlation was between the area of the paretic FDI and PC2 (*p* < 0.01).

Details on the PCA are provided in the Appendix A.

## 4. Discussion

The preliminary results from this proof-of-principle study show that both REMIT and MIT are associated with an increased skill of the paretic upper limb, as indicated by the ul-FM score. Similarities between the two approaches are manifold.

(a)Both MIT and REMIT generate an inter-sensory, visual-proprioceptive conflict.(b)Mirror-evoked sensory conflicts are expected to facilitate the corticospinal output in both treatments [50,51].(c)Both treatments also induce a motor-sensory, not only an inter-sensory, conflict, given that (congruent) visual and proprioceptive feedback is anticipated when the efferent copy of the motor command is released to various cerebral structures [52].(d)In both MIT and REMIT, only one of the two sensory modalities meets the motor expectation from the observed upper limb [53].

However, a main difference exists between MIT and REMIT: in the first model, recovery (if any) stems from action observation coupled with an effort to make proprioception consistent with the visual illusion, while in REMIT, recovery stems from an effort to make vision consistent with proprioception.

Results indicate that both MIT and REMIT improved dexterity, suggesting that the sight of being unable to move the healthy upper limb (REMIT) could effectively promote improvement no less than the sight of being able to move the paretic limb (MIT). During REMIT, no action observation (hence, no mirror neurons) may be claimed to contribute to motor improvements. It might be speculated that a greater effort needs to be generated to visually match the proprioceptive expectation than proprioceptively matching the visual expectation.

This hypothesis aligns with a robust stream of ingenious experiments demonstrating that vision “dominates” over the more reliable proprioception in upper limb voluntary movements [54,55]. The same holds for experiments on standing balance, where vision dominates over the more reliable proprioceptive and vestibular sensations (so-called visual “dependence” or “preference”) [56]. The visual “dominance” over proprioception may develop in many cases of brain stroke when vision areas are spared while somatosensory areas are lesioned [57].

For MIT, the favourable engagement of the mirror neurons system has been hypothesised [58]. Indeed, MIT is associated with the recruitment of the superior temporal gyrus and the premotor cortex, two areas belonging to the mirror neuron system [58]. In humans, action observation alone (even without any mirror illusion) exerts an excitatory action on corticospinal neurons of the primary motor cortex [59]. However, some studies failed to find the recruitment of the mirror neuron system by MIT [60]. On the contrary, these studies pointed to a critical role of multisensory brain areas associated with self-awareness and spatial attention, such as the posterior cingulate cortex and the praecuneus [60]. Improved self-awareness, body representation and attention, three functions strongly related to multisensory interactions, would be responsible for the efficacy (non-specific, indeed) of both MIT [61] and REMIT.

The PCA showed that the post-treatment gains at ul-FM and the area of the cortical map of the healthy FDI muscle correlated with the same PC (i.e., PC1). Correlations had opposite signs. The larger the PC1 component, the smaller the upper limb improvement and the greater the cortical representation of the muscles of the healthy hand. These findings are consistent with the established association between increased excitability of the intact hemisphere and poorer recovery of the paretic hand, which motivates treatments based on non-invasive brain stimulation inhibiting the contralesional motor areas [7].

Systematic reviews and meta-analyses showed that MIT might be an effective treatment to improve upper limb dexterity in selected stroke patients [16,18]. However, these findings are not homogeneous across different investigations [62,63,64]. Therefore, research has been encouraged on more diriment protocols and the design of clinical trials with clear descriptions of the MIT procedure and providing high-quality outcome measures [65]. The current findings, although preliminary, seem original in that they call into question, beyond action observation, the potential role of inter-sensory and effort–feedback congruence. Customisation of treatments is a challenge in rehabilitation, yet, it is the key to success [66]. MIT and REMIT approaches seem complementary. Still, the choice should be tailored to several patients’ characteristics and associated with other exercise or brain stimulation forms.

The present study has limitations. The first and most relevant is the small sample size. The small sample is justified by the need for recruiting stroke patients according to stringent criteria, including a dexterity impairment of modest severity. This flaw heralds low power, although it was possibly attenuated by the double-crossover design. Therefore, other differences between the outcomes of either treatment cannot be ruled out.

Only one patient, P4, suffered from a haemorrhagic stroke, 30 months before this study. The site and size of his lesion were superimposable to those of other two patients (P2 and P7, see Figure 1). These characteristics, along with the rather homogeneous upper limb ability across participants at admission, speak against a possible bias introduced by the type of vascular lesion.

The current study also lacked a stable baseline assessment and a follow-up. Regarding the baseline stability, perhaps the chronicity of the clinical condition warranted it. As for the follow-up, this “proof-of-principle” study had not the ambition to appraise outcome stability.

As just mentioned, this study only enrolled patients with chronic stroke, and an investigation of mirror therapies in acute patients would also be valuable. Indeed both acute and chronic stroke patients can improve their upper limb dexterity, and learning-dependent plasticity can be induced even in the chronic stroke phase, such as through therapeutic exercise [67]. In the present proof-of-concept study we focused on the potential clinical efficacy of the REMIT, as compared to MIT, in chronic stroke in order to control for improvements due to spontaneous recovery.

To complete the previous point, another one that could be raised is that the sample was heterogeneous regarding the months elapsed from the stroke. The shortest and longest stroke to enrolment intervals were 8 and 96 months, respectively.

It must be emphasised again that all the patients recruited here were chronic. Stroke patients are labelled “chronic” usually from six months after the stroke on out [68], with some Authors speaking of chronicity even 60 days after the accident (e.g., [69]). As previously mentioned, what likely matters the most is that acute patients are not mixed with chronic ones.

No information was collected in the current work about the participant’s satisfaction with treatments or their preference for MIT or REMIT. Given the importance of patients’ satisfaction, for example, in determining treatment adherence (e.g., [70]), this information should be collected if MIT and REMIT effectiveness is compared in a clinical trial.

This study did not assess the effectiveness of mirror treatments, compared to many other possible treatments. For instance, these may include hands-on exercises led by a physiotherapist (e.g., Bobath [71] or Kabat methods [72]), forms of EMG biofeedback and/or electrical muscle stimulation [73], forced-use exercises following the learned-nonuse model [74], focal treatments for spasticity with botulinum toxin, non-invasive brain stimulation, etc. In stroke rehabilitation it is customary to associate different treatments, tailored for the single patient and the recovery stage. This “black-box” approach [75] may profit from successful interactions across treatment types, but it makes questionable to compare different “boxes”. The present study only suggests that a “reversed” version of mirror therapy may be a potential ingredient for protocols to be validated.

It should be emphasised that this proof-of-concept study widens the MIT model for the first time. The results indicate that visual-proprioceptive incongruence may elicit some motor recovery of upper limb motion, regardless of action observation.

## 5. Conclusions

Both MIT and REMIT can improve upper limb dexterity in hemiparesis after stroke. The two opposite mirror therapies seem equally effective. This study cues the possible mechanisms of upper-limb motor recovery following mirror treatments. Observation of an illusory movement may force a congruent motor output, but observation of an illusory paresis may also be effective. On these bases, the REMIT approach seems worth investigating in larger samples of stroke patients.

## Figures and Tables

**Figure 1 brainsci-13-00847-f001:**
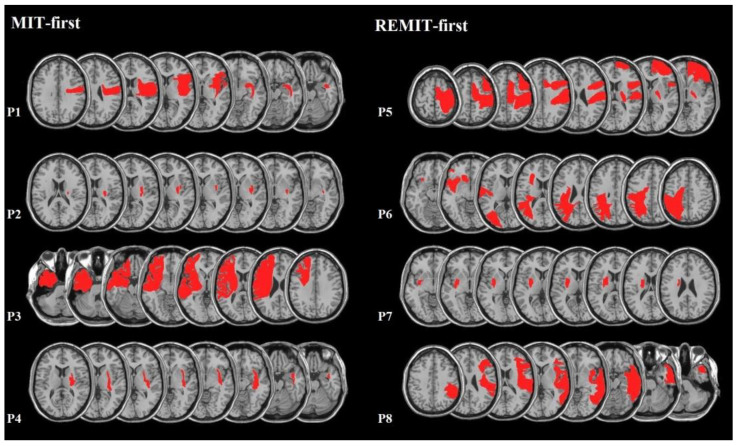
MRI lesion reconstruction for each stroke participant, according to the allocation group. **Left** panel: MIT-first (3 patients with right-sided hemispheric damage); **right** row: REMIT-first (2 patients with right-sided hemispheric damage). The right hemisphere is represented on the reader’s right. Red areas represent the ROI of each patients.

**Figure 2 brainsci-13-00847-f002:**
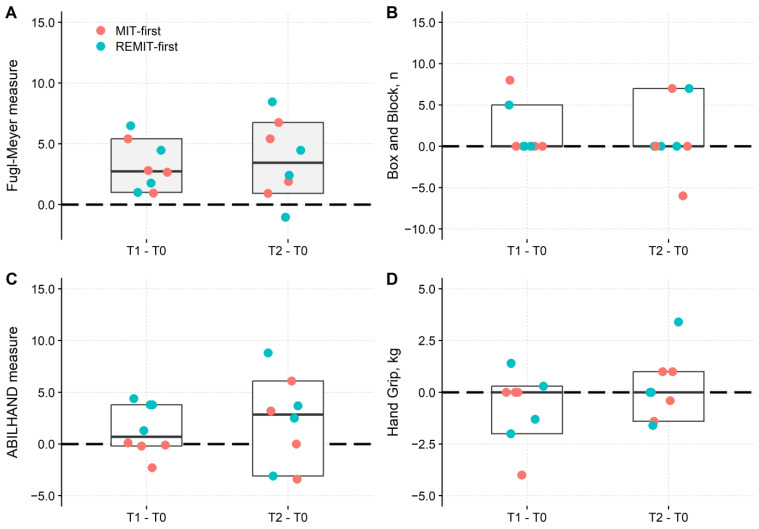
Changes in upper-limb test results after treatments. The four panels show the changes in the Fugl-Meyer upper limb (**A**), Box and Block (**B**), ABILHAND (**C**) and hand grip (**D**), respectively, at each time interval: T1-T0 and T2-T0. In all panels, red dots are the MIT-first group (first week of treatment with MIT). Green dots: REMIT-first group. The boxes report the overall median (horizontal bar) and 95% CI (lower and upper borders). Zero indicates no changes concerning the values recorded at T0 (horizontal dashed line). The difference is given in intervals of 0–100 units for the Fugl-Meyer upper limb and ABILHAND. The number of displaced blocks is reported for the Box and Block test, and the hand grip strength is expressed in Kg. A significant difference compared to T0 was found only for the Fugl-Meyer at both T1 and T2 (grey-filled boxes). Significance testing (i.e., ART-ANOVA) was performed on results collected in the three experimental sessions (i.e., T0, T1 and T2).

**Figure 3 brainsci-13-00847-f003:**
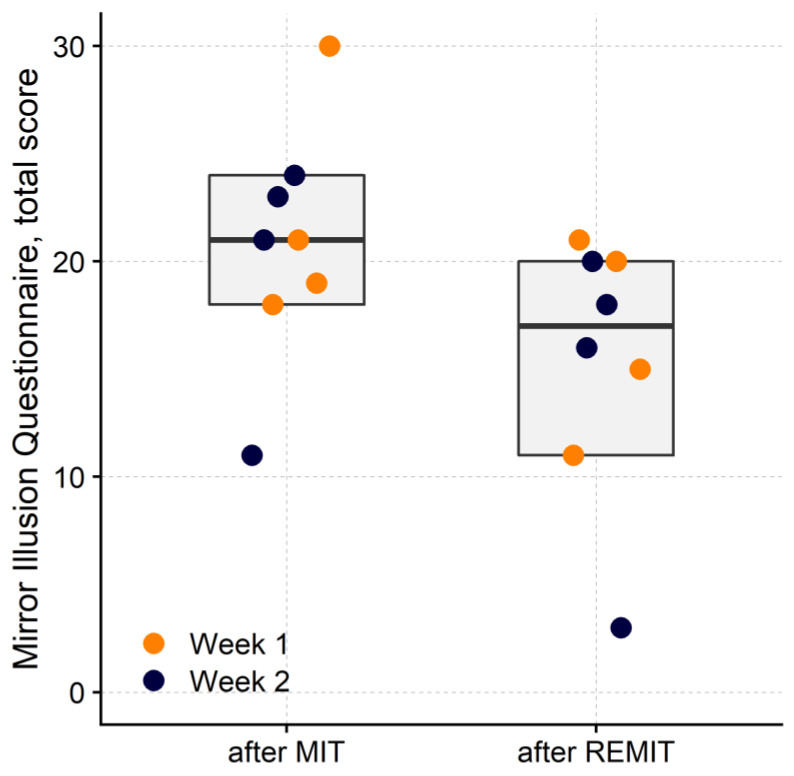
Strength of the mirror illusion. Mirror Illusion Questionnaires (MIQ) were collected at the end of each treatment session. Each dot represents the median questionnaire score (*n* = 5) from a participant in the first (orange) or second (blue) week of treatment after MIT or REMIT. The rectangles show the sample median (horizontal bar) and 95% CI (lower and upper border of the box). After MIT, the total questionnaire score was significantly larger (grey rectangles, *p* < 0.01) than after REMIT.

**Table 1 brainsci-13-00847-t001:** Demographic and clinical characteristics of the stroke sample.

	ID	Age (Years)/ Gender	Education (Years)	Stroke Type	Months from Stroke	Paretic Side	NIH	Bamford Classification	Brunnström Staging	ul-FM	BB	HGS
MIT-First	P1	47, F	13	I	46	L	5	1	4	59.16	32	20
	P2	59, F	13	I	57	L	5	2	4	57.30	16	18
	P3	38, F	16	I	96	R	6	2	3	41.15	4	5
	P4	73, M	17	H	30	L	4	4	3	19.16	0	31
REMIT-First	P5	50, M	18	I	16	L	6	2	2	24.96	0	35.6
	P6	66, M	8	I	27	R	6	2	3	47.46	12	26.6
	P7	77, F	8	I	8	R	4	4	4	54.57	17	15
	P8	70, M	18	I	18	L	3	2	3	36.30	0	18.6

P = participant; M = male, F = female; I = ischaemic stroke, H = haemorrhagic stroke; L = left, R = right; NIH: NIH Stroke Scale. Bamford classification: 1 = total anterior circulation stroke, 2 = partial anterior circulation stroke, 3 = lacunar stroke (no cases observed), 4 = posterior circulation stroke. Pre-treatment baseline scores for each participant on tests used to assess MIT and REMIT are also given. ul-FM = Fugl-Meyer assessment of the upper limb; BB = Box and Block test; HGS = hand grip strength. Interval measures of the ul-FM and ABILHAND are given on a 0–100 scale (the higher, the better). All patients were right-handed.

**Table 2 brainsci-13-00847-t002:** Motor cortical excitability data of patients.

		RMT (uV)		MEPs (mV)		Area	
		AH	UH	AH	UH	AH	UH
MIT-first	P1	45	52	0.2	0.8	9	16
	P2	56	40	0.4	0.5	15	7
	P3	/	57	0.0	0.8	0	10
	P4	/	44	0.0	1	0	8
REMIT-first	P5	/	55	0.0	0.7	0	10
	P6	/	39	0.0	2.4	0	13
	P7	90	41	0.2	0.8	17	3
	P8	77	65	0.4	3	7	14

RMT = resting motor threshold (uV); MEPs = motor evoked potentials measured from first dorsal interosseous (FDI); Area: sum of the total excitable sites (1.5 cm grid cells size); AH = affected hemisphere; UH = unaffected hemisphere. MIT was the first of two treatments in patients P1 to P4.

**Table 3 brainsci-13-00847-t003:** Statistical analysis of the behavioural outcome measures.

	Fugl-Meyer Upper Limb			Box and Block Test			Hand Grip Strength			ABILHAND		
	F	Df	*p*-Value	F	Df	*p*-Value	F	Df	*p*-Value	F	Df	*p*-Value
Session	6.23	2.12	0.01	0.93	2.12	0.42	0.47	2.12	0.64	0.74	2.12	0.50
Group	0.03	1.6	0.87	0.49	1.6	0.51	0.48	1.6	0.51	1.39	1.6	0.28
Session × Group	0.54	2.12	0.60	1.15	2.12	0.35	0.037	2.12	0.96	1.50	2.12	0.26

## Data Availability

The raw data supporting the conclusions of this article will be made available by the authors without undue reservation (Zenodo repository).

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
