# Peer review of "Reversed Mirror Therapy (REMIT) after Stroke—A Proof-of-Concept Study"

_brainsci, 2023, doi:10.3390/brainsci13060847_

Round 1

Reviewer 1 Report

This study was designed to investigate the effect of MIT and REMIT in patients with stroke. As a result, both MIT and REMIT were similar effects in patients with stroke. This method was interesting. However, there were some points to explain in more detail.

Page 1, line 32

Please add references.

Page 2, line74

The methods and mechanisms of MIT were described. However, you should add the effect of MIT on patients with stroke.

Page 2, line 91

The sample size is too small. Does it meet the sample size requirements for this study? If so, please add a formula to calculate the sample size. If not, then it should be converted to a preliminary study, etc.

Page 3, line 113

What is method of randomization ?

Page 3, line 116

Are these patients receiving rehabilitation in addition to MIT? If so, add the content and amount of rehabilitation. 

Page 3, line 124

These require further detailed information. In particular, please add more details about the MRI imaging setup. 

Nothing.

Author Response

see attachment

Comments and Suggestions for Authors

This study was designed to investigate the effect of MIT and REMIT in patients with stroke. As a result, both MIT and REMIT were similar effects in patients with stroke. This method was interesting. However, there were some points to explain in more detail.

We thank the Reviewer for this positive comment.

Page 1, line 32. Please add references.

Two references were added.

  • Upper limb recovery after stroke: The stroke survivors' perspective [1];
  • Accelerating Stroke Recovery: Body Structures and Functions, Activities, Participation, and Quality of Life Outcomes From a Large Rehabilitation Trial [2].

Note that to comply with a request from Reviewer 3, a recent reference from 2018 was also included in addition to a classical one.

Page 2, line 74. The methods and mechanisms of MIT were described. However, you should add the effect of MIT on patients with stroke.

MIT's effect on stroke was quickly mentioned in the previous Introduction version (line 64).

Nowadays, a growing body of evidence suggests that MIT could be effective in selected stroke patients in improving upper limb functioning. This aspect is reported more clearly in the revised Introduction to comply with the Reviewer's request (lines 66-68).

Note that this addition is relatively condensed since details on what is asked by the Reviewer 1 is provided in the Discussion (lines 482-486).  In addition, we preferred not to overturn the Introduction not to counteract the positive comment from Reviewer 2 ("Introduction perfectly reflects the aim of the study").

Page 2, line 91. The sample size is too small. Does it meet the sample size requirements for this study? If so, please add a formula to calculate the sample size. If not, then it should be converted to a preliminary study, etc.

We fully agree with Reviewer 1 that the sample size is small, and our findings should be considered preliminary. This aspect was discussed among the study's limitations in the previous manuscript (see lines 468-472 of the new version).

Reviewer 1 highlights that this should be classified as a preliminary study in this comment. Again, because of the reduced sample size, Reviewers 2 and 3 ask to classify it as a pilot study.

As stated in the Methods section (page 2, line 86), we considered the current study a proof-of-concept. This fact is also stressed in the Discussion's beginning and end (lines 409 and 480, respectively).

The "proof-of-concept" label was used here according to the definition used in trials for drug development. In this field, a proof-of-concept trial is a trial indicating that the treatment under investigation "might work" in the intended therapeutic area [3]. This definition of the proof of concept closely resembles that of a trial run to show preliminary efficacy.

Different definitions of pilot studies have been put forward.

Similarly to a proof-of-concept, a pilot study is "a small-scale test of the methods and procedures to be used on a larger scale if the pilot study demonstrates that these methods and procedures can work" [4].

Other sources [5] elaborated on the definition above to stress that a pilot study aims not to test hypotheses about an intervention's effectiveness but to assess treatment feasibility in a more extensive study. According to this definition, showing that "procedures can work" or that the treatment "under investigation might work" are not the aims of a pilot study.

Because of the ambiguities about the pilot study definition, we prefer to keep the proof-of-concept label instead of the pilot one.

However, to comply with Reviewer 1 request, the title has been modified as follows: "Reversed Mirror Therapy (REMIT) after stroke. A proof-of-concept study". Proof-of-concept has been added to the title to stress the preliminary nature of the findings reported here.

In addition, in the revised manuscript, our results are defined as "preliminary" at the beginning of the Discussion, too (line 409).

Page 3, line 113. What is method of randomisation?

Information added (coin tossing, line 131).

Page 3, line 116. Are these patients receiving rehabilitation in addition to MIT? If so, add the content and amount of rehabilitation.

Mirror therapies were administered alone and not associated with other rehabilitative treatments. The main text now clarified this aspect (Methods, paragraph 2.2, lines 129-130).

Page 3, line 124. These require further detailed information. In particular, please add more details about the MRI imaging setup.

The MRI exam was a standard structural MRI scan done just for clinical purposes. We have only manually reconstructed the lesion site and size as now detailed at page 7, lines 315-322.

Comments on the Quality of English Language. Nothing.

Ok. 

References

1        Barker RN, Brauer SG. Upper limb recovery after stroke: the stroke survivors’ perspective. Disabil Rehabil 2005;27:1213–23. doi:10.1080/09638280500075717

2        Lewthwaite R, Winstein CJ, Lane CJ, et al. Accelerating Stroke Recovery: Body Structures and Functions, Activities, Participation, and Quality of Life Outcomes From a Large Rehabilitation Trial. Neurorehabil Neural Repair 2018;32:150–65. doi:10.1177/1545968318760726

3        Al-Shurbaji A. Proof-of-Principle/Proof-of-Concept Trials in Drug Development. In: Pharmaceutical Sciences Encyclopedia. John Wiley & Sons, Ltd 2010. 1–17. doi:10.1002/9780470571224.pse251

4        Porta M, editor. A Dictionary of Epidemiology. In: A Dictionary of Epidemiology. Oxford University Press 2014. https:// (accessed 3 May 2023).

5        Pilot Studies: Common Uses and Misuses. NCCIH. https://www.nccih.nih.gov/grants/pilot-studies-common-uses-and-misuses (accessed 3 May 2023).

Reviewer 2 Report

Manuscript shows the Mirror and Reversed Mirror Therapy for upper limb paresis after stroke. I have some considerations for you.

Introduction perfectly reflects the aim of the study.

Methodology.

Why did you choose 8 patients? Is there any ample size calculation? If it isn’t the number of participants should be placed at results section.

How long the recruitment took part? Who implement the recruitment?

Were sessions individualized or group? Did participants follow another treatment or these sessions are the only treatment these days?

Have participants any experience with mirror therapy before? Did they receive any instructions or informative sheet about the procedure?

Who assess participants, the same physiotherapist who applied the treatment or another person?

Was fatigue taken into account when performing the treatment?

I have seen that exercises are in the annex but, could a mention or brief summary of them be added?

Results.

8 patients participated in the study and 7 of them are ischemic, could the hemorrhagic one, affect the results?

Discussion

As author know acute and chronic stroke patients don´t have the same limitations. Some information that support your choice “chronic” instead acute patients must be explained at the discussion section.

Besides, there are several physiotherapy techniques to treat these patients. A comparison between these ones and MIT and REMIT could be added.

I miss some results about the patient’s experience of sensation about the treatment. Were participants satisfied with the treatment? did they have any preference in favor or against both treatments?

Regarding the sample size, maybe “a pilot study” should be added to the title.

Funding code should be added.

Author Response

see attachment

Comments and Suggestions for Authors

Manuscript shows the Mirror and Reversed Mirror Therapy for upper limb paresis after stroke. I have some considerations for you.

Introduction perfectly reflects the aim of the study.

Thank you for this positive comment.

Methodology.

Why did you choose 8 patients? Is there any ample size calculation? If it isn't the number of participants should be placed at results section.

The sample size was not calculated because  this  is a proof of concept, preliminary study (see also our reply to Reviewer 1 and our answer to another comment from Reviewer 2 below).

As requested by Reviewer 2, the total number of participants is reported for the first time in the Results section (lines 299-300), instead of Methods.

How long the recruitment took part? Who implement the recruitment?

Due to the very stringent inclusion criteria, recruitment lasted two years. Patients were recruited by one of the Authors (GF). This information has been added to the manuscript (line 97).

Were sessions individualised or group? Did participants follow another treatment or these sessions are the only treatment these days?

Patients received one-on-one sessions (i.e. one therapist treated one patient at a time).

As explained in reply to Reviewer 1, participants only received mirror therapies.

Both of these clarifications were added in the Methods section 2.2 (lines 125-130).

Have participants any experience with mirror therapy before? Did they receive any instructions or informative sheet about the procedure?

Participants were naïve to mirror therapies (they had never experienced them before; information added to the Methods, line 97).

No informative sheet was provided (lines 164-166).

Who assess participants, the same physiotherapist who applied the treatment or another person?

No, the therapist who administered the treatment did not assess patients.

This aspect was explained at the beginning of Section 2.2 of the previous manuscript: "Both MIT and REMIT were administered to all patients in 5 daily one-on-one sessions (Monday to Friday) led by a trained physiotherapist engaged in neither patient assessment nor data analysis".

The paragraph has been rearranged to highlight this point (lines 125-128).

Was fatigue taken into account when performing the treatment?

As customary with therapeutic exercise, physical and cognitive fatigue was considered during mirror therapies administering. Fatigue was not of concern during treatments, even if this side effect was not recorded in the study with dedicated measures.

This clarification has been added to the end of Section 2.2 (lines 167-169).   

I have seen that exercises are in the annex but, could a mention or brief summary of them be added?

The list of the 19 exercises, with a brief description for each exercise, has been added to Section 2.2 (lines 140-158).

Results.

8 patients participated in the study and 7 of them are ischemic, could the hemorrhagic one, affect the results?

In our opinion it appears unlikely that including a single chronic patient who suffered a haemorrhagic stroke could bias the results. First, the stroke occurred 30 months before the patient's enrolment; thus, no bleeding was present at the study onset. In addition, the lesion was comparable to that of two other (ischaemic) patients (P2 and P7), in terms of site and size.

However, these points have been added to the first paragraphs of the Results section (lines 325-330; see also Discussion 473-477).

Discussion

As author know acute and chronic stroke patients don't have the same limitations. Some information that support your choice "chronic" instead acute patients must be explained at the discussion section.

Thank you for this suggestion.

A paragraph discussing this issue from Reviewer 2 has been added to the Discussion end in the sections reporting the study's limitations (lines 482-488).

Besides, there are several physiotherapy techniques to treat these patients. A comparison between these ones and MIT and REMIT could be added.

A paragraph detailing the different physiotherapy techniques to treat upper limb impairment in stroke has been added to the Discussion (501-510).

I miss some results about the patient's experience of sensation about the treatment. Were participants satisfied with the treatment? did they have any preference in favor or against both treatments?

We are sorry, but this information is missing. A note about this has been added to the study's limitations (497-500).

Regarding the sample size, maybe "a pilot study" should be added to the title.
Reviewers 1 and 3 also raised a similar issue. As a result, the title was changed to the following after considering the three Reviewers' indications: " Reversed Mirror Therapy (REMIT) after stroke. A proof-of-concept study".

Funding code should be added.

Funding has been better specified as follows: "This research was funded by the Italian Ministry of Health (Ricerca Corrente, IRCCS Istituto Auxologico Italiano)", which is ok for our funder.

References

1        Barker RN, Brauer SG. Upper limb recovery after stroke: the stroke survivors’ perspective. Disabil Rehabil 2005;27:1213–23. doi:10.1080/09638280500075717

2        Lewthwaite R, Winstein CJ, Lane CJ, et al. Accelerating Stroke Recovery: Body Structures and Functions, Activities, Participation, and Quality of Life Outcomes From a Large Rehabilitation Trial. Neurorehabil Neural Repair 2018;32:150–65. doi:10.1177/1545968318760726

3        Al-Shurbaji A. Proof-of-Principle/Proof-of-Concept Trials in Drug Development. In: Pharmaceutical Sciences Encyclopedia. John Wiley & Sons, Ltd 2010. 1–17. doi:10.1002/9780470571224.pse251

4        Porta M, editor. A Dictionary of Epidemiology. In: A Dictionary of Epidemiology. Oxford University Press 2014. https:// (accessed 3 May 2023).

5        Pilot Studies: Common Uses and Misuses. NCCIH. https://www.nccih.nih.gov/grants/pilot-studies-common-uses-and-misuses (accessed 3 May 2023).

6        States RA, Pappas E, Salem Y. Overground physical therapy gait training for chronic stroke patients with mobility deficits. Cochrane Database Syst Rev 2009;2009:CD006075. doi:10.1002/14651858.CD006075.pub2

7        Ward NS, Brander F, Kelly K. Intensive upper limb neurorehabilitation in chronic stroke: outcomes from the Queen Square programme. J Neurol Neurosurg Psychiatry 2019;90:498–506. doi:10.1136/jnnp-2018-319954

8        Caronni A, Ramella M, Arcuri P, et al. The Rasch Analysis Shows Poor Construct Validity and Low Reliability of the Quebec User Evaluation of Satisfaction with Assistive Technology 2.0 (QUEST 2.0) Questionnaire. Int J Environ Res Public Health 2023;20:1036. doi:10.3390/ijerph20021036

9        Murakami Y, Honaga K, Kono H, et al. New Artificial Intelligence-Integrated Electromyography-Driven Robot Hand for Upper Extremity Rehabilitation of Patients With Stroke: A Randomized, Controlled Trial. Neurorehabil Neural Repair 2023;:15459683231166940. doi:10.1177/15459683231166939

10       Swayne OBC, Rothwell JC, Ward NS, et al. Stages of Motor Output Reorganization after Hemispheric Stroke Suggested by Longitudinal Studies of Cortical Physiology. Cerebral Cortex 2008;18:1909–22. doi:10.1093/cercor/bhm218

11       Kim SY, Allred RP, Adkins DL, et al. Experience with the “good” limb induces aberrant synaptic plasticity in the perilesion cortex after stroke. J Neurosci 2015;35:8604–10. doi:10.1523/JNEUROSCI.0829-15.2015

12       Rizzolatti G, Craighero L. The mirror-neuron system. Annu Rev Neurosci 2004;27:169–92. doi:10.1146/annurev.neuro.27.070203.144230

Reviewer 3 Report

The subject of this work seems interesting to the readers, however there are several aspects to take into account before its possible publication.

First of all the title is very long, they should change it and make it more attractive.

In the inclusion criteria I do not understand why the age of the participants has to be between 50 and 80 years old, could you clarify it for me?

There are two inclusion criteria with the number 4, the authors must review it and make the corresponding changes.

Where were the participants recruited and by what method? They must include this information.

One of the inclusion criteria is to be over 50 years old, however, MIT therapy patient number 1 is 47 years old and number 3 is 38, how was that possible?

The difference in terms of months of evolution of the stroke is too high between different patients, there are patients of 8 months and others of 96 months of evolution... it is too great

the bibliographical references used should be more current, some are many years old.

The sample is more of a pilot study than a randomized clinical trial.

The language needs small changes and necessary adjustments of some terms

Author Response

see attachment

Comments and Suggestions for Authors

The subject of this work seems interesting to the readers, however there are several aspects to take into account before its possible publication.

First of all the title is very long, they should change it and make it more attractive.

This Reviewer 3's comment contradicts a comment from Reviewer 2, who asks for more details in the title.

In any case, the title was changed with a more concise one: "Reversed Mirror Therapy (REMIT) after stroke. A proof-of-concept study".

In the inclusion criteria I do not understand why the age of the participants has to be between 50 and 80 years old, could you clarify it for me?

Please accept our apologies for this typo. There were no reasons to restrict so much the age window.  Actually, the age inclusion criterion was between 35 and 80 years.

There are two inclusion criteria with the number 4, the authors must review it and make the corresponding changes.

Thank you for noticing this inaccuracy which was amended.

Where were the participants recruited and by what method? They must include this information.

Participants were consecutively recruited, as reported in the Methods section of the previous manuscript version (opening lines of section 2.1 in the Methods). In the revised manuscript, the sentence was rearranged to improve readability, and this aspect has also been reported in the Methods beginning (lines 100-102). 

Patient enrolment took place at the Department of Neurorehabilitation Sciences of a large Teaching Hospital in Milan, Italy. Patients who received rehabilitative treatments in our Department in the years before the study onset were contacted and offered to participate. This information is now provided at the beginning of the new Methods section (lines 100-102). 

One of the inclusion criteria is to be over 50 years old, however, MIT therapy patient number 1 is 47 years old and number 3 is 38, how was that possible?

There was a typo when reporting the inclusion criteria (see our answer to a previous Reviewer's comment), which has been solved in the revised manuscript.

The difference in terms of months of evolution of the stroke is too high between different patients, there are patients of 8 months and others of 96 months of evolution... it is too great

This issue from Reviewer 3 has also been discussed among the study's limitations (lines 489-496).

We do not think that gathering patients from 8 to 96 months after stroke causes a bias.
According to the literature  all these patients can be labelled as "chronic" [1]. For sure, secondary  impairments (e.g., muscle contractions and hypotrophy) might emerge even years after storke. In this study, however, the important point was that the present mobility on the paretic upper limb was rather homogeneous across subjects.

the bibliographical references used should be more current, some are many years old.

Four new references from recent years (2018 [2], 2019 [3] and two from 2023 [4,5]) have been added. We left “old” references because they seem to  represent bibliographic milestones on the topic (e.g. [6–8]).

The sample is more of a pilot study than a randomised clinical trial.

A similar point was also raised by Reviewers 1 and 2. Accordingly, to comply with the reviewers' request, the title has been changed to stress the preliminary nature of the findings reported here since the beginning of the manuscript. In addition, in the revised manuscript, our results are also defined as "preliminary" in the Discussion opening.

We agree that the sample size is small, and our findings should be considered preliminary. This aspect was discussed among the study's limitations in the previous manuscript.

As stated in the Methods section (line 91), we considered the current study a proof-of-concept. This fact is also stressed in the Discussion's beginning and end (line 409 and 511, respectively).

The "proof-of-concept" label was used here according to the definition used in trials for drug development. In this field, a proof of concept trial is a trial that indicates that the compound under investigation "might work" in the intended therapeutic area [9]. This definition of the proof of concept closely resembles that of a trial run to show preliminary (rather than definite) efficacy.

Reviewer 3 suggests specifying that our study is a pilot one. However, different definitions of pilot studies have been put forward.

Similarly to a proof-of-concept, a pilot study is "a small-scale test of the methods and procedures to be used on a larger scale if the pilot study demonstrates that these methods and procedures can work" [10].

Other sources [11] elaborated on the definition above to stress that a pilot study aims not to test hypotheses about an intervention's effectiveness but to assess treatment feasibility in a more extensive study. According to this definition, showing that "procedures can work" or that the treatment "under investigation might work" are not the aims of a pilot study.

Because of the ambiguities about the pilot study definition, we prefer to keep the proof-of-concept label instead of the pilot one.

Comments on the Quality of English Language

The language needs small changes and necessary adjustments of some terms

We refined the English verbiage.

References

1        States RA, Pappas E, Salem Y. Overground physical therapy gait training for chronic stroke patients with mobility deficits. Cochrane Database Syst Rev 2009;2009:CD006075. doi:10.1002/14651858.CD006075.pub2

2        Lewthwaite R, Winstein CJ, Lane CJ, et al. Accelerating Stroke Recovery: Body Structures and Functions, Activities, Participation, and Quality of Life Outcomes From a Large Rehabilitation Trial. Neurorehabil Neural Repair 2018;32:150–65. doi:10.1177/1545968318760726

3        Ward NS, Brander F, Kelly K. Intensive upper limb neurorehabilitation in chronic stroke: outcomes from the Queen Square programme. J Neurol Neurosurg Psychiatry 2019;90:498–506. doi:10.1136/jnnp-2018-319954

4        Caronni A, Ramella M, Arcuri P, et al. The Rasch Analysis Shows Poor Construct Validity and Low Reliability of the Quebec User Evaluation of Satisfaction with Assistive Technology 2.0 (QUEST 2.0) Questionnaire. Int J Environ Res Public Health 2023;20:1036. doi:10.3390/ijerph20021036

5        Murakami Y, Honaga K, Kono H, et al. New Artificial Intelligence-Integrated Electromyography-Driven Robot Hand for Upper Extremity Rehabilitation of Patients With Stroke: A Randomized, Controlled Trial. Neurorehabil Neural Repair 2023;:15459683231166940. doi:10.1177/15459683231166939

6        Swayne OBC, Rothwell JC, Ward NS, et al. Stages of Motor Output Reorganization after Hemispheric Stroke Suggested by Longitudinal Studies of Cortical Physiology. Cerebral Cortex 2008;18:1909–22. doi:10.1093/cercor/bhm218

7        Kim SY, Allred RP, Adkins DL, et al. Experience with the “good” limb induces aberrant synaptic plasticity in the perilesion cortex after stroke. J Neurosci 2015;35:8604–10. doi:10.1523/JNEUROSCI.0829-15.2015

8        Rizzolatti G, Craighero L. The mirror-neuron system. Annu Rev Neurosci 2004;27:169–92. doi:10.1146/annurev.neuro.27.070203.144230

9        Al-Shurbaji A. Proof-of-Principle/Proof-of-Concept Trials in Drug Development. In: Pharmaceutical Sciences Encyclopedia. John Wiley & Sons, Ltd 2010. 1–17. doi:10.1002/9780470571224.pse251

10       Porta M, editor. A Dictionary of Epidemiology. In: A Dictionary of Epidemiology. Oxford University Press 2014. https:// (accessed 3 May 2023).

11       Pilot Studies: Common Uses and Misuses. NCCIH. https://www.nccih.nih.gov/grants/pilot-studies-common-uses-and-misuses (accessed 3 May 2023).

Round 2

Reviewer 2 Report

Dear Authors,

The title is much better. Thank you for considering my suggestions.

For future studies try to collect a bigger sample size.